# miR-1285-3p targets TPI1 to regulate the glycolysis metabolism signaling pathway of Tibetan sheep Sertoli cells

Xuejiao An[1,2], Taotao Li[1,2], Nana Chen[1,2], Huihui Wang[1,2], Manchun Su[1,2], Huibin Shi[1,2], Xinming Duan[3], Youji Ma[1,2]*

1 College of Animal Science and Technology, Gansu Agricultural University, Lanzhou, China, 2 Gansu Key Laboratory of Animal Generational Physiology and Reproductive Regulation, Lanzhou, China, 3 Nongfayuan (Zhejiang) Agricultural Development Co., Ltd., Huzhou, Zhejiang, China

* yjma@gsau.edu.cn

**Data Availability Statement:** The original data was stored in the Dryad repository at: https://doi.org/10.5061/dryad.5dv41ns7k (DOI: 10.5061/dryad.5dv41ns7k).

## Abstract

Glycolysis in Sertoli cells (SCs) can provide energy substrates for the development of spermatogenic cells. Triose phosphate isomerase 1 (TPI1) is one of the key catalytic enzymes involved in glycolysis. However, the biological function of TPI1 in SCs and its role in glycolytic metabolic pathways are poorly understood. On the basis of a previous research, we isolated primary SCs from Tibetan sheep, and overexpressed *TPI1* gene to determine its effect on the proliferation, glycolysis, and apoptosis of SCs. Secondly, we investigated the relationship between *TPI1* and miR-1285-3p, and whether miR-1285-3p regulates the proliferation and apoptosis of SCs, and participates in glycolysis by targeting *TPI1*. Results showed that overexpression of *TPI1* increased the proliferation rate and decreased apoptosis of SCs. In addition, overexpression of *TPI1* altered glycolysis and metabolism signaling pathways and significantly increased amount of the final product lactic acid. Further analysis showed that miR-1285-3p inhibited *TPI1* by directly targeting its 3'untranslated region. Overexpression of miR-1285-3p suppressed the proliferation of SCs, and this effect was partially reversed by restoration of *TPI1* expression. In summary, this study shows that the miR-1285-3p/TPI1 axis regulates glycolysis in SCs. These findings add to our understanding on the regulation of spermatogenesis in sheep and other mammals.

## 1. Introduction

Tibetan sheep (*Ovis aries*), one of the three major coarse-haired sheep breeds in China, is the most populous domestic animal on the Qinghai-Tibet Plateau and its adjacent areas located 3000 m above sea level [1]. It provides extremely important living materials and economic income for local farmers and herdsmen. However, its population is severely restricted largely due to the reproductive physiological characteristics of Tibetan sheep, such as long development cycle, low fecundity, and late sexual maturity. At present, the molecular biology involving testis development and spermatogenesis of male Tibetan sheep has not been fully elucidated. As the most critical organ in maintaining the reproductive ability of male animals,

**Funding:** The study was supported by the Education science and technology innovation project of Gansu Province (GSSYLXM-02), National Natural Science Foundation of China (31960662), National Key R&D Program of China (2021YFD1100502) and "Innovation Star" project for outstanding graduate students of the Education Department of Gansu Province (2021CXZX-350). The funders had no role in study design, data collection and analysis, decision to publish, or preparation of the manuscript.

**Competing interests:** The authors declare that they have no known competing financial interests or personal relationships that could have appeared to influence the work reported in this paper.

the main biological function of the testis is to produce sperm [2]. Studies have revealed that the process of spermatogenesis is regulated by many genes at multiple levels, such as epigenetics, transcription, and translation [3]. Therefore, exploring the expression characteristics and regulation of related genes in Tibetan sheep testicular development and spermatogenesis has important scientific significance for understanding the molecular mechanism through which maintenance of testicular function in sheep and other plateau animals is regulated.

In the process of spermatogenesis, Sertoli cells (SCs) can not only provide physical structure scaffolds and the microenvironment required for survival of spermatogenic cells with different degrees of differentiation in the seminiferous tubules, but also provide various levels of spermatogenic cells for energy production [4]. A previous study reported that SCs can be closely connected with adjacent seminiferous tubules to form a blood-testis barrier which prevents invasion by certain exogenous toxic substances, thereby playing an extremely important role in spermatogenesis [5]. Simultaneously, the lactic acid produced by SCs during glycolytic metabolism pathways can be used as an energy substrate for the development of spermatogenic cells [6]. It could also promote RNA in spermatogenic cells and protein synthesis, thereby reducing apoptosis of spermatogenic cells [7]. Glycolytic metabolism in SCs means that, under the action of glucose transporters on the cell membrane of SCs, sugars such as starch and sucrose are passively transported into the cytoplasm of SCs by the glucose produced by catabolism. In addition, acetone is produced through catalysis of a series of enzymes. The enzymes are then converted into lactic acid, which ultimately enters the spermatogenic cells [8–10]. Among them, triosephosphate isomerase 1 (TPI1), one of the key enzyme molecules in the glycolytic pathway, can catalyze the conversion of dihydroxyacetone phosphate to glyceraldehyde-3-phosphate [11].

It is worth noting that TPI1 was formed before eukaryotic/prokaryotic differentiation, and it was one of the older enzyme families [12]. The lack of TPI1 enzyme in mammals might cause hemolytic anemia, and even death of individual organisms in severe cases [13]. In higher animals, mutation of the *TPI1* gene can cause a rare autosomal recessive genetic disease referred to as TPI1 deficiency [14]. Studies in mice found that the *TPI1* gene is expressed in spermatogenic cells in the testis and the main segment of the sperm flagella in the epididymis [15]. On the other hand, studies in humans have revealed that anti-sperm antibodies can target *TPI1*, thereby preventing sperm acrosome reaction and secondary binding of sperm to the zona pellucida [16]. Moreover, *in vitro* experiments found that the addition of *TPI1* inhibitors can significantly reduce sperm motility in rats [17]. The above findings suggest that the *TPI1* gene might have the function of positively regulating the development of spermatogenic cells. In our earlier studies [18], we explored the proteomics of sheep testis tissue and found that the TPI1 protein was expressed in sheep testis tissue. After conducting further functional annotations, we found that the TPI1 protein participates in the glycolytic metabolism pathway. At the same time, we cloned the *TPI1* gene in the testis tissue of Tibetan sheep at different developmental stages (before pre-puberty, sexual maturity, and adult), and explored its expression and location in the testis and epididymis tissues. Results showed that the Tibetan sheep *TPI1* gene CDS 833th Synonymous mutations (C→T) occur in the bases, their mRNA and protein expressions increase with age, and it was mainly located in SCs [19]. However, the specific role of the *TPI1* gene in the glycolytic metabolism of SCs, and even in testicular development and spermatogenesis has not yet been reported in mammals, especially domestic animals.

MicroRNAs (miRs, miRNAs) are small noncoding single-stranded RNAs that lack protein-encoding functions. It has been reported that miRNAs can bind to the 3′ untranslated regions (3′-UTRs) of their target mRNAs to suppress protein expression [20]. To date, many studies have found that miRNAs modulate the progression of most human cancers [21]. MiR-1285-3p, with a length of 22 nt and located on chromosome 7, was the first miRNA to be discovered

in human embryonic stem cells. One study found that it could specifically bind to p53 and exert an inhibitory effect on tumor cells by regulating the expression level of p53 [22]. Notably, miR-1285-3p is a highly conserved miRNA, which is aberrantly expressed in a variety of tumor cells. A recent study reported that it mainly regulates downstream genes, and inhibits proliferation, migration, and invasion of tumor cells, thereby exerting a tumor suppressor effect [23]. In addition, studies in breast cancer have predicted that nine miRNAs, including miR-1285-3p, target TPI1 (the key enzyme in glycolysis), and produce anti-glycolysis and anti-proliferation effects [24]. Herein, we used the miRDB prediction software (http://www.mirdb. org) to predict the target miRNAs of the *TPI1* gene. Results indicated that there were six miR-NAs that had a targeting relationship with *TPI1*. Furthermore, their expression in Tibetan sheep SCs was verified by real-time quantitative polymerase chain reaction (RT-qPCR). Among them, the expression level of miR-1285-3p was the highest. However, only a handful of studies have explored the role of miR-1285-3p in glycolysis metabolism, and it is not yet clear whether it targets *TPI1* and plays a role in the glycolysis of SCs.

In the present study, we isolated Tibetan sheep primary SCs, and effectively transfected them with *TPI1* overexpression vector and siRNA silencing vector, with the overarching goal of exploring the effects of *TPI1* gene overexpression and silencing on the proliferation and apoptosis of SCs, as well as the influence of glycolytic metabolic pathways. Based on the obtained results, we discovered that *TPI1* was inhibited by miR-1285-3p and could reverse the inhibitory effect of miR-1285-3p on the glycolytic metabolism of SCs. Elucidation of the underlying mechanism found that the glycolytic metabolism signaling pathway was regulated by the miR-1258-3p/TPI1 axis, and further verified that miR-1285-3p inhibits the proliferation of SCs by targeting *TPI1*. Collectively, our findings reveal the key role of miR-1285-3p/TPI1 axis in mediating the activation of the SCs glycolytic metabolism signaling pathway.

## 2. Materials and methods

### 2.1. Isolation and culture of primary sheep SCs

All animal experiments were performed in accordance with the animal care and experimental procedure guidelines approved by the Animal Committee of Gansu Agricultural University (GSAU-AEW-2020-0057). Three well-grown 3-month-old Tibetan sheep were selected and brought back to the laboratory for intravenous injection of sodium pentobarbital, no heartbeat, continuous involuntary breathing for 2–3 min, no blink reflex, and then bilateral testes tissues were collected. SCs cells were isolated according to a protocol described by Xuejiao *et al.* [25]. Briefly, three well-grown 3-month-old Tibetan sheep were selected and brought back to the laboratory. The animals were first anaesthetized with intravenous injection of sodium pentobarbital to ensure that there was no heartbeat, no blink reflex, and there was continuous involuntary breathing for 2–3 min. Next, bilateral testes tissues were isolated, sterilized using 75% alcohol, and then the surface envelope was cut (the aseptic part should be taken out with scissors as much as possible). The tissues were digested with trypsin and collagenase, followed by addition of serum-containing DMEM/F12 (Gibco, New York, USA) to stop the digestion and screening with different mesh cells. Cells were then cultured in 15% fetal bovine serum (FBS, Gibco, New York, USA) supplemented with 1% penicillin streptomycin sol in DMEM/F12 (FBS, Gibco, New York, USA). After the cells were adhered and differentiated, they were collected, and discontinuous density gradient centrifugation was performed to purify the cells using Percoll cell separation solution (Solarbio, Beijing, China) with volume fractions of 11%, 19%, 27%, 35%, and 43%, respectively. The differential adhesion method was then used to purify cells several times in order to obtain relatively pure SCs. Notably, the

specific GATA4 antibody was used to identify SCs by immunofluorescence staining. When the concentration reached more than 90%, the cells could be used for subsequent tests.

## 2.2. Construction of TPI1 gene silencing or overexpression vectors

The TPI1-siRNA interference sequence was designed according to our earlier cloned Tibetan sheep *TPI1* gene sequence (MN847717), and the negative control (NC-siRNA) was used as the control group (S1 Table). Notably, both the TPI1-siRNA and NC-siRNA vectors were synthesized by Genepharma Biological. On the other hand, the *TPI1* overexpression vector (pc-DNA-3.1(+)-TPI1) was constructed by Genewiz Biological, and the empty vector (pc-DNA-3.1(+) was used as the control group. The plasmid small extraction kit (Tiangen, Beijing, China) was used to extract the plasmid according to the manufacturer's instructions, and then the recombinant plasmid was double digested with *NheI* and *NotI* restriction enzymes. Finally, the digestion products were identified by 2% agarose gel electrophoresis.

## 2.3 The dual-luciferase reporter gene

The Target Scanprediction software (https://www.targetscan.org) was used to predict the miRNA targeted by the *TPI1* gene. It was found that the 56–61 region of the *TPI1* 3'UTR had a binding site with miR-1285-3p. Primers were then designed according to the 3'-UTR sequence of *TPI1*, followed by synthesis of *TPI1* 3'-UTR wild-type gene fragments and mutant gene fragments. It should be noted that the mutant gene fragment is a sequence that mutates the binding point sequence of *TPI1* to the reverse complementary sequence of the original target binding point. Next, pmiRGLO was used as a vector to synthesize plasmids of *TPI1* 3'-UTR wild-type gene fragment (TPI1 3'UTR WT) and mutant gene fragments (TPI1 3'UTR MUT). MiR-1285-3p mimics, NC mimics, miR-1285-3p inhibitor, and NC inhibitor were also designed and synthesized (Genepharma Biological, Shanghai, China). The synthetic sequences are shown in S2 Table. The TPI1 3'UTR WT and TPI1 3'UTR MUT plasmids were combined with miR-1285-3p mimics, NC mimics, miR-1285-3p inhibitor, and NC inhibitor NC, followed by transfection into HEK293T cells. Finally, the luciferase activity was measured 48 h after transfection using a dual-luciferase assay system (Promega, Beijing, China).

## 2.4. Cell transfection

SCs were selected of the F5 generation after isolated, purified and identified, the transfection only began when the cells reached 60%-70% confluence. On one hand, 2500 ng of the extracted *TPI1* overexpression plasmid DNA was diluted with 250μL serum-free medium Opti-MEM (Gibco, New York, USA), and gently mixed, whereas on the other hand, 5μL Lipofectamine 2000 (Invitrogen, California, USA) was diluted with 250μL serum-free medium Opti-MEM and gently mixed gently, followed by incubation at room temperature for 5 min. The two liquids were then mixed and incubated at room temperature for 20 min to form a DNA-Lipofectamine 2000 complex. Next, 500μL of the complex was taken and added to 1.5 mL of Opti-MEM in a basic six-well plate. The two liquids were gently mixed, placed in a 37˚C incubator for 4 h, and then replaced with DMEM/F12 medium supplemented with 15% FBS for culture. Notably, the dosage of siRNA for transfection was 200nM/well, the dosage of miR-1285-3p mimics and NC mimics was 50nM/well, the dosage of miR-1285-3p inhibitor and NC inhibitor was 100nM/well, and the lipofectamine 2000 dosage was 5μL/well. The rest of the steps were similar to the steps involved in the transfection of plasmids.

## 2.5. Cell counting kit-8 (CCK-8) proliferation assay

CCK-8 reagent (MedChen Express, Shanghai China) was used to determine cell viability. Briefly, SCs growing in the logarithmic phase were selected, digested with trypsin, and counted. Cells were then seeded in a 96-well plate at a density of $1 \times 10^4$ cells/well for 24 h, followed by transfection for 24, 48, and 72 h. Next, 10μL of CCK-8 reagent was added to each well and incubated for 1 h at 37°C. Finally, the OD value of each well was measured at 450 nm using enzyme-linked immunosorbent assay (ELISA) and the growth curve was plotted.

## 2.6. Flow cytometry

Flow cytometry was used to detect Annexin V-FITC/PI double-labeled necrotic and apoptotic cells (Q2 + Q3). Briefly, cells were digested with EDTA-free trypsin to make a single cell suspension, and washed with pre-cooled PBS. Next, 300μL of the binding buffer was added to the cells at a density of about $5 \times 10^5$ cells/tube, followed by addition of 5μL of Annexin V-FITC in each tube, and thorough mixing. 5μL of PI solution was then added to the tube, followed by thorough mixing and protection from light for 10 min. Finally, the liquid in the tube was detected using flow cytometry.

## 2.7. Total RNA extraction, cDNA synthesis, and RT-qPCR analysis

The total RNA of SCs after transfection were extracted using TransZol (TransGen Biotech, Beijing, China) according to manufacturer's instructions. The integrity of RNA was determined by 1% agarose gel electrophoresis, whereas the concentrations and quality of RNA samples were examined by NanoDrop 2000 (Thermo Fisher Scientific, Waltham, MA, USA)) and Agilent 2100 (Agilent, Santa Clara, CA, USA), respectively. Next, *Evo M-MLV* RT Kit with gDNA Clean for RT-qPCR (Accurate Biotech, Hunan, China) was used to reverse transcribe the RNA into cDNA, and SYBR Green Premix *Pro Taq* HS qPCR Kit (Accurate Biotech, Hunan, China) was used to perform qPCR on the Roche LightCycler96 in accordance with the manufacturer's protocols. Relative mRNA expression was normalized to β-actin (a housekeeping gene) mRNA and calculated using the $2^{-\Delta\Delta Ct}$ method [26]. Reverse transcription of miRNA using mir-XTM miRNA first-strand Synthesis Kit (Takara, Japan), the qPCR method was the same as that of mRNA. S3 and S4 Tables shows the primers used for RT-qPCR analysis.

## 2.8. Western blot analysis

After transfection of SCs, they were homogenized and lysed using a radioimmunoprecipitation assay (RIPA) protein extraction kit (Solarbio, Beijing, China) according to the manufacturer's instructions. Protein concentrations were then quantified using a commercial bicinchoninic acid (BCA) protein assay (Solarbio, Beijing, China). Next, the extracted proteins were denatured with 4x protein loading buffer (DTT, Solarbio, Beijing, China), resolved using 12% sodium dodecyl sulfate polyacrylamide gel electrophoresis (SDS-PAGE), and then transferred onto polyvinylidene difluoride (PVDF) blotting membranes (Beyotime, Shanghai, China). Membranes were first blocked with phosphate buffered saline tween-20 (PBST) containing 5% non-fat milk for 2 h at room temperature, followed by incubation with either rabbit anti-TPI1 polyclonal antibody (1:600; Bioss, Beijing, China) or anti-beta-actin polyclonal antibody (1:600; Bioss, Beijing, China) at 4°C overnight. On the next day, membranes were washed with PBST, and then incubated with goat anti-rabbit IgG/HRP antibody (1:5000; Bioss, Beijing, China) for 2 h at 37°C. Finally, they were washed with PBST and the protein signals were visualized using NcmECL Ultra reagents (New Cell & Molecular Biotech Co.LTD, Suzhou, China) in an X-ray room.

## 2.9. Detection of the final products and content of key enzymes of the glycolytic metabolic signal pathway

After transfection, the cell fluid was collected and the lactic acid extraction kit (Nanjing Jiancheng Bioengineer Institute, Nanjing, China) was used to measure the content of lactic acid production during glycolysis process. In addition, the SCs were collected after transfection, and then the ATP kit (Nanjing Jiancheng Bioengineer Institute, Nanjing, China), lactate dehydrogenase (LDH) kit (Nanjing Jiancheng Bioengineer Institute, Nanjing, China), and pyruvate extraction kit (Nanjing Jiancheng Bioengineer Institute, Nanjing, China) were used to measure the change level of key enzymes, energy, and pyruvate content in the glycolysis process. Notably, all the experiments were conducted following the manufacturer's instructions.

## 2.10. Statistical analyses

All statistical analyses were performed using SPSS 21.0 software and all data were expressed as mean ± standard error (Mean±SE). The gray value of the band obtained after western blot analysis was scanned and determined using AlphaEaseFC image analysis software. The relative expression of the target protein was the ratio of the gray value of the target protein to the gray value of β-actin. One-way analysis of variance (ANOVA) was used to compare differences among multiple groups. $P<0.05$ was considered statistically significant.

# 3. Results

## 3.1 Identification of Tibetan sheep primary SCs

Immunofluorescence staining of SCs with specific antibody GATA4 and TPI1 antibody showed that purified SCs could synthesize and express SCs-specific binding protein GATA4, and staining with TPI1 antibody showed that TPI1 protein was expressed in SCs. The results could rule out contamination by other testis cell types, and the purified SCs could be used in subsequent experiments (Fig 1).

## 3.2 Recombinant plasmid double digestion and determination of transfection efficiency

The contents obtained after double digestion of pcDNA3.1(+)-PGAM1 were electrophoresed on a 2% agarose gel (Fig 1F). Results showed that the size of the digestion product was consistent with the theoretical value. QRT-PCR was then performed to determine the expression level of *TPI1* mRNA in the transfected sheep SCs. The results revealed that the expression of *TPI1* gene in SCs of the si-TPI1-1, si-TPI1-2, and si-TPI1-3 transfection groups were all downregulated after transfection for different time periods compared to the control group, but the silencing effect was most obvious after transfection for 48 h (Fig 2A–2C). The transfection efficiency of the three siRNAs was then compared after transfection for 48 h. It was found that the silencing effect of si-TPI1-1 was significantly higher than the others ($P<0.05$) (Fig 2D), indicating that transfection of si-TPI1-1 for 48 h had the best silencing effect. Compared to the empty vector group, the expression of *TPI1* gene in the pcDNA3.1(+)-TPI1 transfection group was upregulated after transfection for different time periods. However, the upregulation was the most obvious after 48 h, indicating that the overexpression efficiency was the best after transfection for 48 h (Fig 2E).

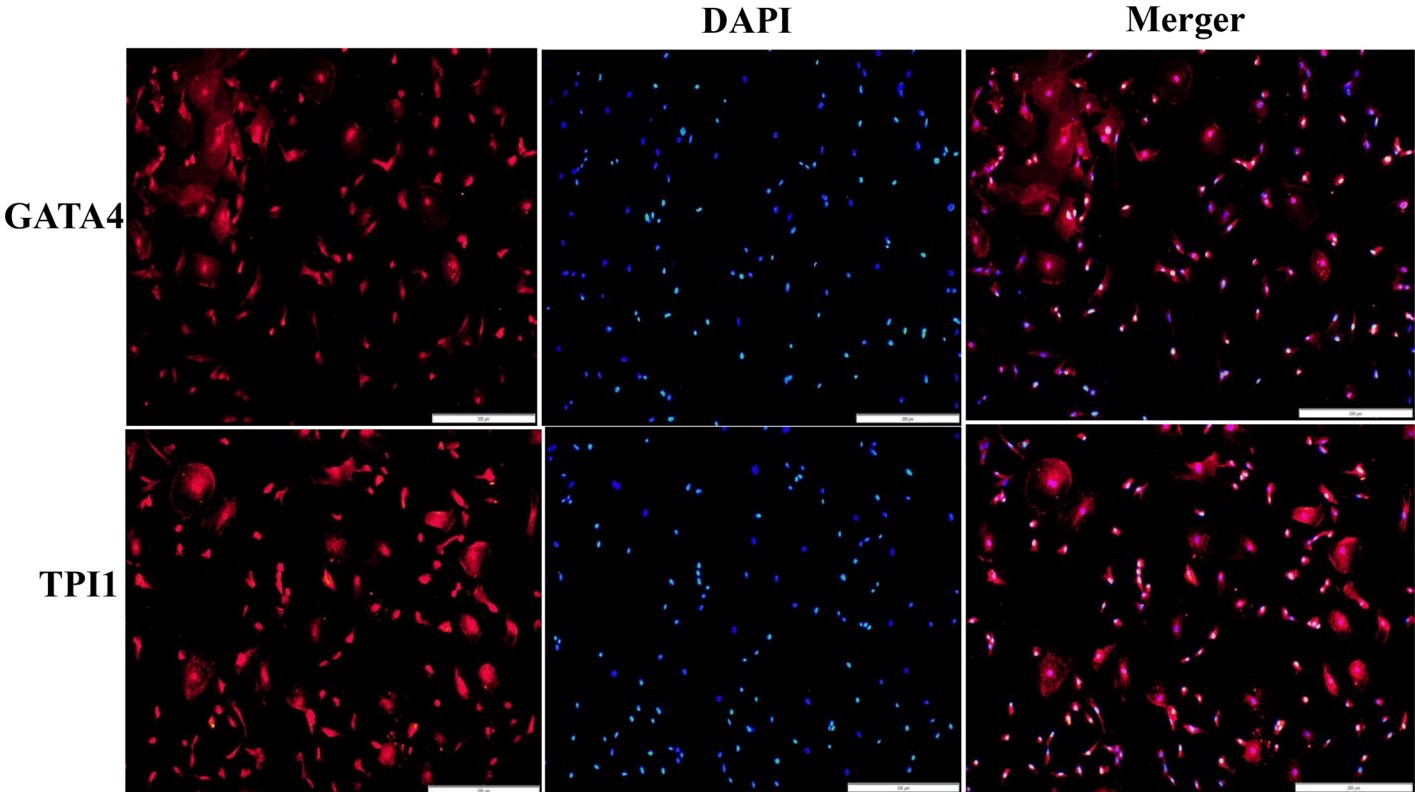

**Fig 1. Immunofluorescence staining of primary SCs of Tibetan sheep (20×) [25].**

### 3.3 Western blot analysis of TPI1 protein expression

The si-TPI1 plasmid with the highest interference efficiency and its control NC, as well as the overexpression plasmid pcDNA3.1(+)-TPI1 and the empty vector were transfected into sheep SCs for 48 h. Western blot analysis was then performed to determine the expression of TPI1 protein. Results showed that pcDNA3.1(+)-TPI1 could significantly increase the expression of TPI1 compared to the control group ($P<0.05$), and si-TPI1 could significantly reduce the expression of TPI1 compared to the control group ($P<0.05$) (Fig 3). These results suggest that pcDNA3.1(+)-TPI1 could effectively promote the expression of TPI1 protein in sheep SCs, whereas si-TPI1 could effectively inhibit the expression of TPI1 protein in sheep SCs.

### 3.4 Effect of TPI1 gene silencing or overexpression on the proliferation, apoptosis, and cycle of SCs

The proliferation of SCs after transfection was determined by CCK-8 assay and flow cytometry. The CCK-8 test results showed that there was no significant difference in overexpression or silencing after 24h of transfection ($P>0.05$). However, the number of cells in the pcDNA3.1 (+)-TPI1 group was significantly higher than that in the empty vector group after 48 h and 72 h ($P<0.05$). In addition, the number of cells in the si-TPI1 group was significantly lower than that in the NC group ($P<0.05$) (Fig 4A and 4C). Flow cytometry was used to determine the changes in cell apoptosis after 48 h of transfection. Results showed that the apoptosis rate of the pcDNA3.1(+)-TPI1 group was significantly lower than that of the empty vector group whether in early or late apoptosis ($P<0.05$). The apoptosis rate of the si-TPI1 group was higher than that of the NC group ($P<0.05$) (Fig 4B). Furthermore, RT-qPCR was used to determine

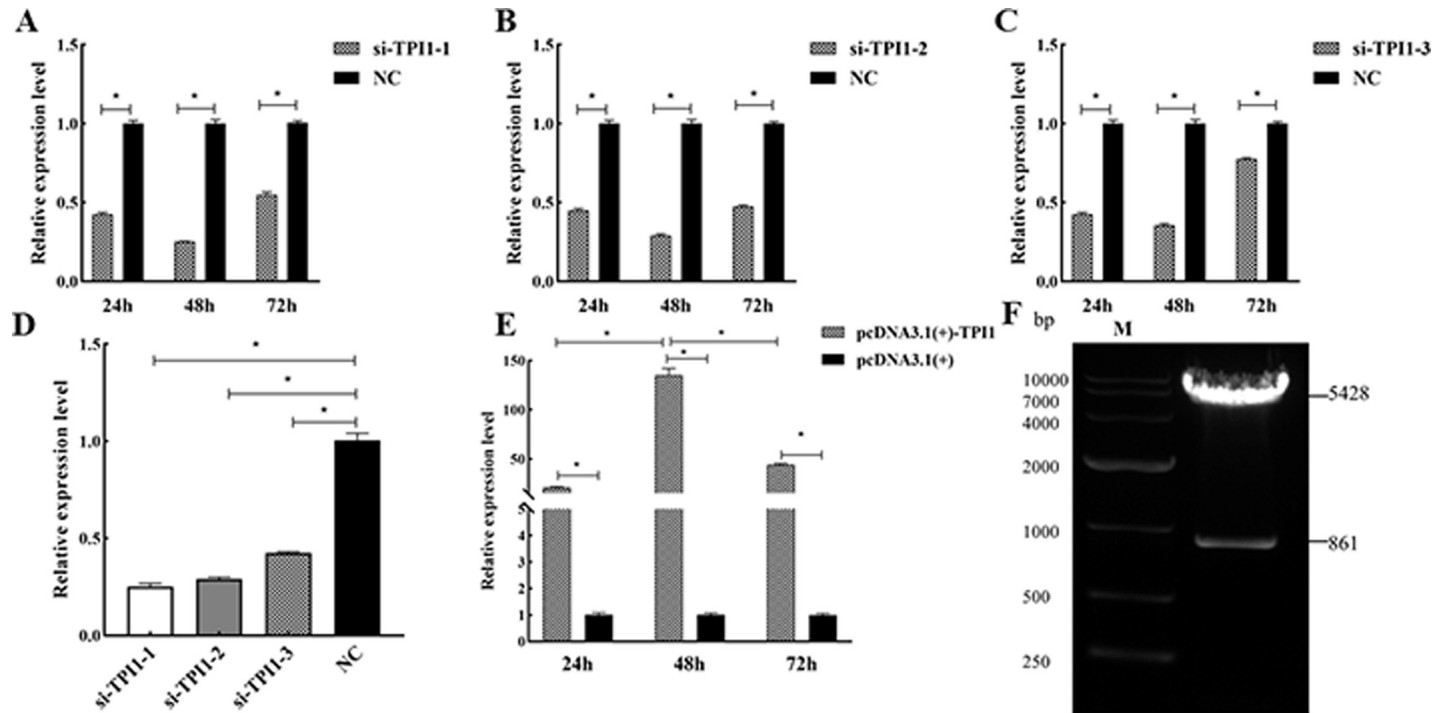

**Fig 2. Evaluation of recombinant plasmid and detection of transfection efficiency.** (A-E) *TPI1* gene silencing or overexpression efficiency. (F) Evaluation of recombination plasmids. "∗" indicate significant difference (*P*<0.05), the same below.

the changes in the expression of proliferation-, cycle-, and apoptosis-related genes at the mRNA level. The results demonstrated that the relative expression of pro-apoptotic genes (*Caspase3* and *Bax*) was significantly downregulated in the overexpression group compared to the empty vector group (*P*<0.05), whereas the expressions of proliferation-related genes (*PCNA* and *Bcl2*) and cell cycle-related gene (*CyclinB1*) were significantly upregulated in the overexpression group compared to the empty vector group (*P*<0.05). After silencing the *TPI1* gene, the results were in contrast to the overexpression results (Fig 4D and 4E). Collectively, these results suggest that the overexpression of *TPI1* inhibited cell apoptosis, whereas silencing of *TPI1* promoted cell apoptosis.

### 3.5 Effect of silencing or overexpression of TPI1 on the glycolytic metabolism pathway

After 48 h of transfection, the expression levels of the eight genes downstream of the *TPI1* gene in the glycolytic metabolism pathway in the SCs was determined by qRT-PCR. The results

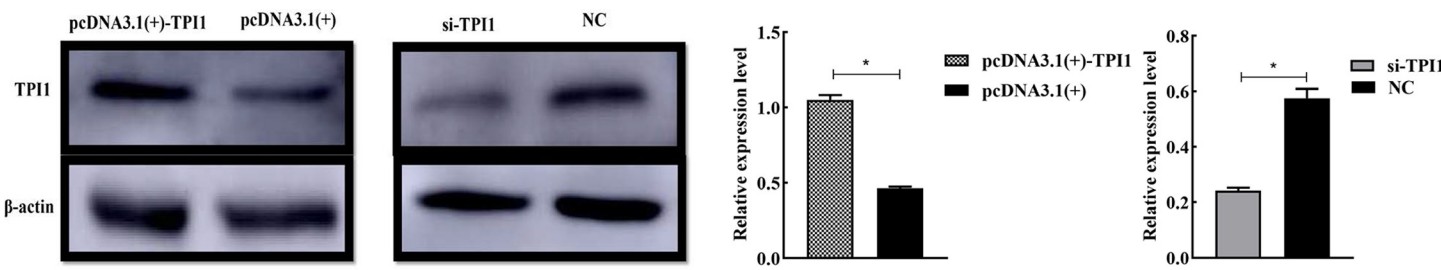

**Fig 3. Expression patterns of TPI1 protein in SCs after silencing or overexpression.**

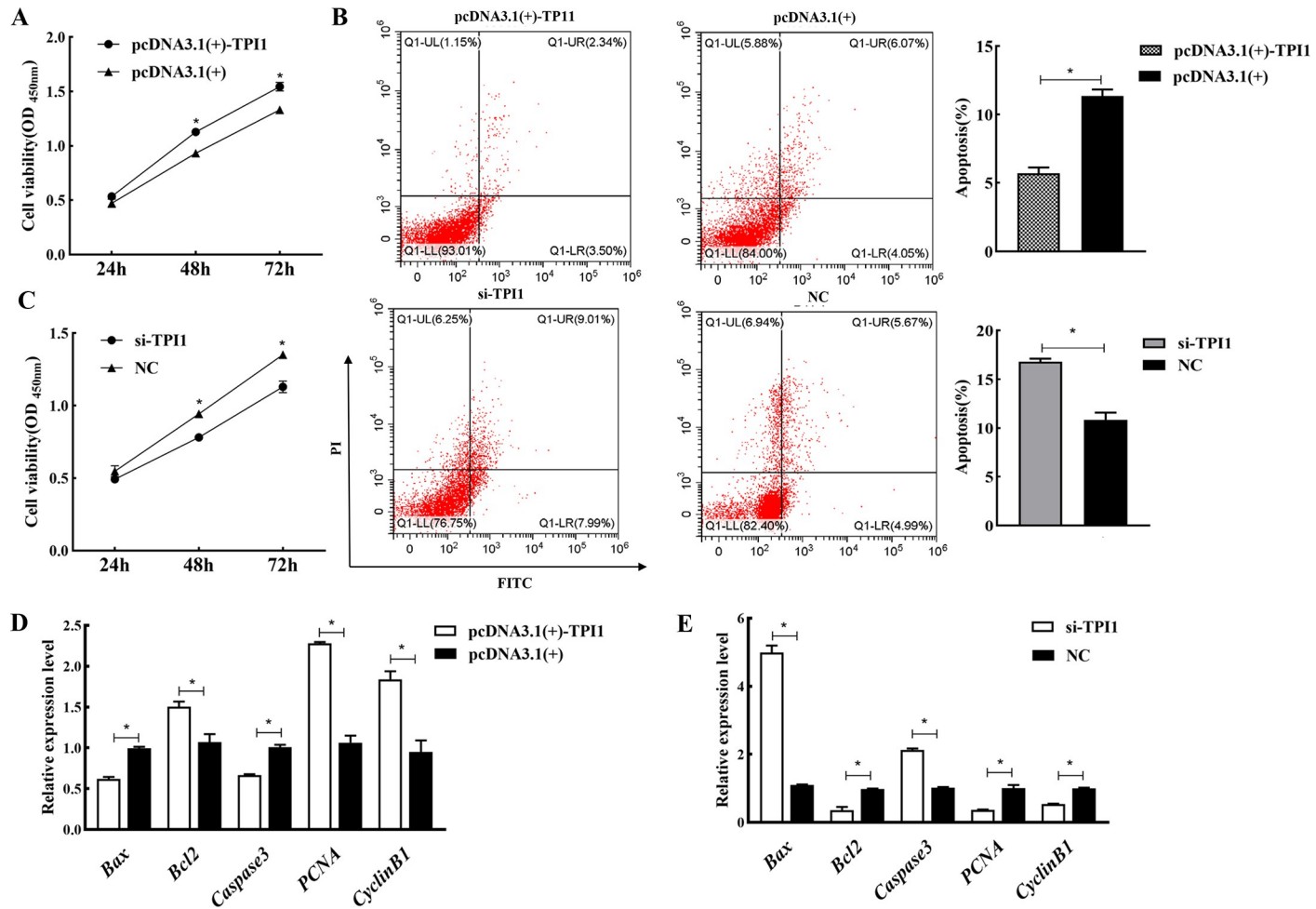

**Fig 4. Effect of *TPI* gene silencing and overexpression on the proliferation, cycle and apoptosis of SCs.** A, C: CCK-8 detects the cell proliferation rate after overexpression and silencing of *TPI1*; B: Flow cytometry was used to detect the apoptosis of SCs after overexpression and silencing of *TPI1*; D, E: The effect of overexpression and silencing of *TPI1* gene on the expression of proliferation, cycle and apoptosis related genes.

showed that the expression levels of *ENO1*, *PKM*, *LDHA*, *LDHB*, *MCT1*, *PGAM1*, and *PGK* genes were significantly upregulated in the pcDNA3.1(+)-TPI1 group compared to the empty vector group ($P<0.05$), whereas the expressions of *ENO1*, *PKM*, *LDHA*, *LDHB*, *MCT1*, *GAPDH*, and *PGK* genes were significantly downregulated in the si-TPI1 group compared to the NC group ($P<0.05$). This suggested that overexpression of the *TPI1* gene could increase the expression of downstream genes, whereas silencing of the *TPI1* gene could reduce the expression of downstream genes (Fig 5A and 5C). The pyruvate, lactic acid, and ATP kits were then used to determine the content of pyruvate and ATP in SCs, and the activity of LDH after 48 h of transfection. Results revealed that the LDH activity, lactic acid production, and contents of pyruvate and ATP in the pcDNA3.1(+) group were significantly higher than those in the pcDNA3.1(+)-TPI1 group ($P<0.05$). After silencing the *TPI1* gene, the result was the opposite of overexpression (Fig 5B, 5D, 5E and 5F).

### 3.6 TPI1 is a direct target of miR-1285-3p

We studied the expression of 6 miRNAs predicted to be targeted to TPI1 in SCs and found that the expression level of miR-1285-3p in SCs was significantly higher than that of other

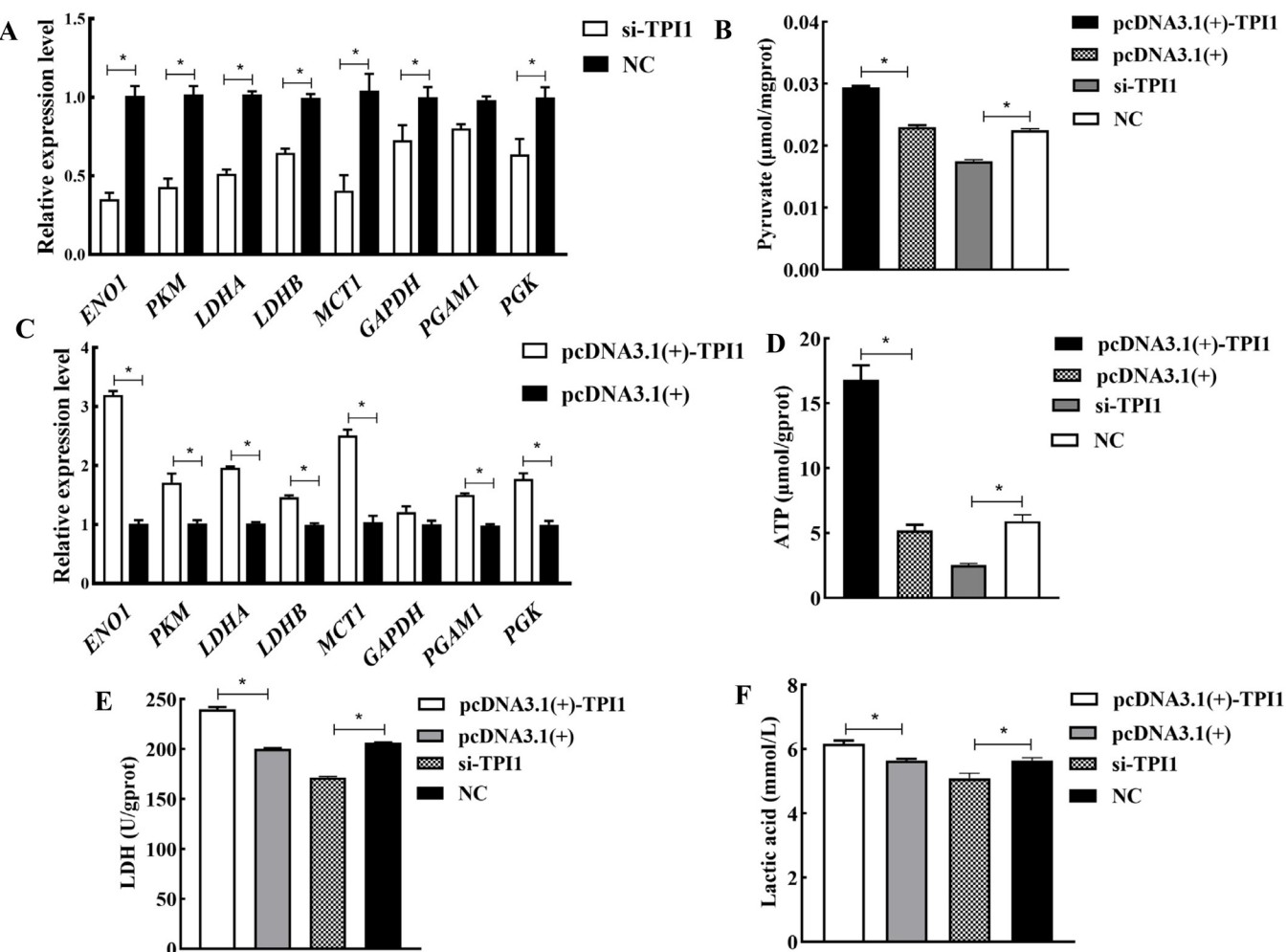

**Fig 5. Effect of *TPI1* gene silencing and overexpression on glycolytic metabolic pathways.** A, C: The expression levels of downstream genes in the glycolytic metabolic pathway after silencing and overexpression of the *TPI1* gene. B, D, E and F: The content of glycolytic metabolic pathway products, energy and key enzymes after silencing and overexpression of *TPI1* gene.

miRNAs(*P*<0.05) (Fig 6F). We first confirmed the interaction between miR-1285-3p and *TPI1* with the predicted sites (Fig 6A). The dual-luciferase assay system indicated that the miR-1285-3p mimics significantly decreased the luciferase activity of TPI1-WT, but not TPI1-MUT (Fig 6B). Moreover, it was found that knockdown of miR-1285-3p significantly increased TPI1 expression, whereas enforced expression of miR-1285-3p significantly decreased TPI1 expression in SCs (Fig 6D and 6H). These results suggest that miR-1285-3p can inhibit the expression of *TPI1* by directly binding to the 3'-UTR of TPI1. In addition, the results of CCK8 assay, flow cytometry, and the expression level of apoptosis-related and proliferation-related genes showed that the overexpression of miR-1285-3p could inhibit the proliferation of SCs (Fig 6C, 6E and 6G). The expression abundance of the eight genes downstream of the *TPI1* gene in the glycolytic metabolism pathway was determined by qRT-PCR after transfection of the SCs. Results showed that the expression of these genes was significantly reduced in miR-1285-3p mimics compared to the negative control (*P*<0.05), whereas their expression was significantly increased in the miR-1285-3p inhibitor group compared to the negative control (*P*<0.05) (Fig 6J). Next, ELISA kits were used to detect LDH activity, lactic

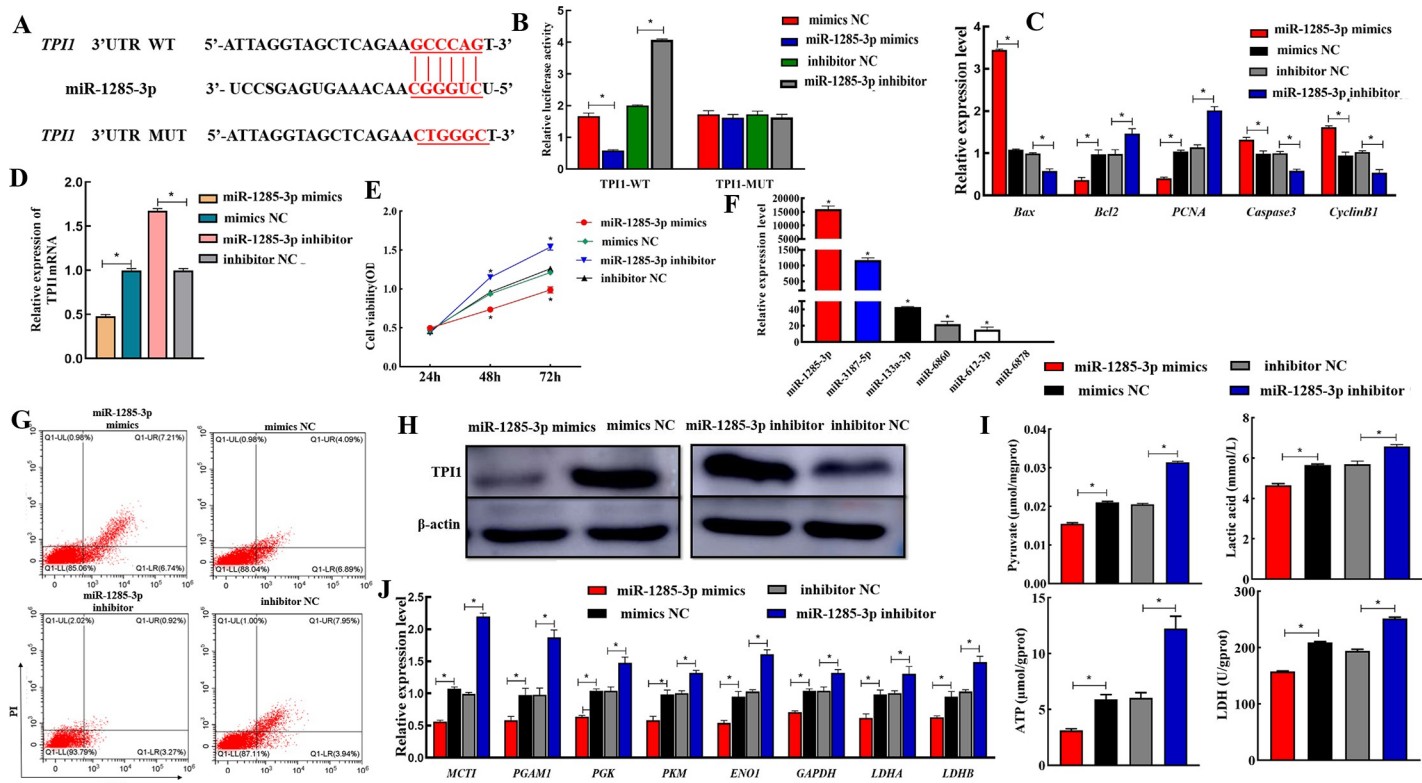

**Fig 6. TPI1 expression is downregulated by miR-1285-3p directly targeting of the 3'-UTR of TPI1.** A, TPI1 was found for the potential regulatory targets of miR-1285-3p using prediction tool. B, Dual-luciferase reporter assay analysis of the target relationship of miR-1285-3p with TPI1. F, Expression of miRNA in SCs. The TPI1 expression levels were determined by real-time PCR analysis (D) western blot (H) after transfection with the miR-1285-3p mimics or negative control or after transfection with the miR-1285-3p inhibitor or negative control in SCs. CCK-8 (E), Flow cytometry (G) and the expression of proliferation, cycle and apoptosis related genes (C) was used to detect the apoptosis of SCs after transfection miR-1285-3p mimics, negative control, miR-1285-3p inhibitor or negative control. The expression levels of downstream genes (J) and the content of glycolytic metabolic pathway products, energy and key enzymes (I) of SCs after transfection miR-1285-3p mimics, negative control, miR-1285-3p inhibitor or negative control.

acid production, and content of pyruvate and ATP during glycolysis in SCs. It was found that their contents were significantly lower in the miR-3614-5p mimics compared to the negative control group ($P<0.05$), and significantly increased in the miR-1285-3p inhibitor group compared to the negative control group ($P<0.05$) (Fig 6I). Overall, these results suggest that *TPI1* is a direct functional target of miR-1285-3p in the glycolytic metabolic pathway of SCs.

### 3.7 Rescue experiment verified that miR-1285-3p negatively regulates SCs glycolytic metabolism by targeting TPI1

Given the potential suppressive role of miR-1285-3p in SCs glycolytic metabolism, we further explored whether *TPI1* mediated the suppressive role of miR-1285-5p in SCs. QRT-PCR and western blot results confirmed that co-transfection of miR-1285-3p with the *TPI1* overexpression plasmid mildly rescued the TPI1 expression in SCs (Fig 7A and 7B). Further functional experiments showed that restoring the expression of *TPI1* can effectively reverse the decline in the proliferation of SCs caused by overexpression of miR-1285-3p (Fig 7C, 7D and 7F). Furthermore, the expression of glycolytic metabolism signaling pathway-related molecules was decreased after enforced expression of miR-1285-3p in SCs, whereas reintroduction of *TPI1* abolished the suppressive effect of miR-1285-3p mimics on the glycolytic metabolism signaling pathway (Fig 7A, 7E and 7G). In addition, given that *TPI1* is an important enzyme in the

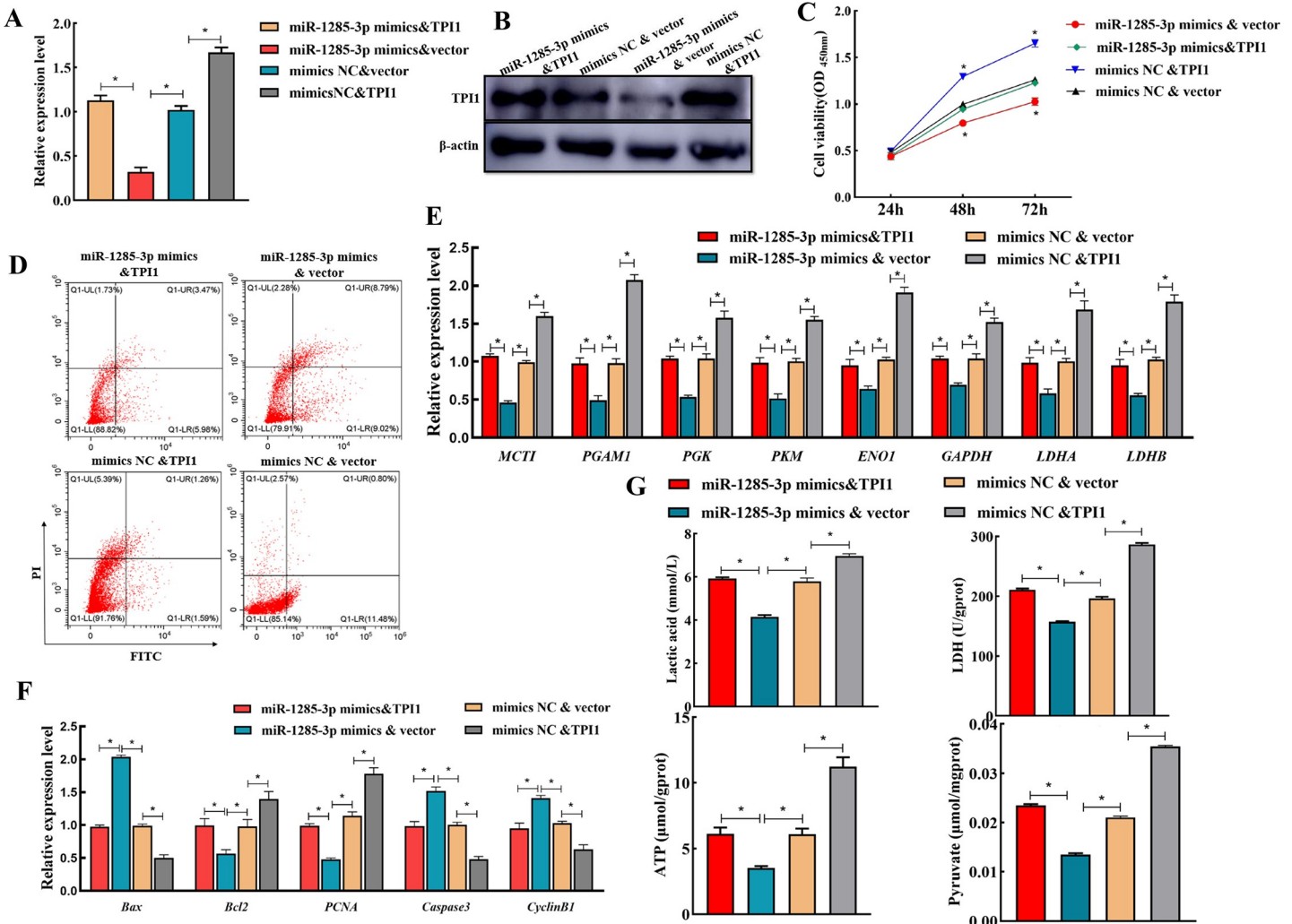

**Fig 7. Effect of miR-1285-3p on SCs glycolytic metabolism by targeting TPI1 in rescue experiment.** A, B: RNA and protein level of TPI1 were detected by qRT-PCR and western blot, respectively. C, D and F: SCs proliferation and apoptosis detected by CCK-8, Flow cytometry and the expression of proliferation, cycle and apoptosis related genes. E, G: The expression levels of downstream genes and the content of glycolytic metabolic pathway products, energy and key enzymes of SCs.

aerobic glycolysis process, we further confirmed the effect of miR-1285-3p/TPI1 axis on SCs glucose metabolism. Altogether, these findings suggest that the miR-1285-3p/TPI1 axis promotes SCs progression, at least in part, via the glycolytic metabolism signaling pathway.

## 4. Discussion

As one of the key enzyme molecules in the glycolysis pathway, TPI1 catalyzes conversion of dihydroxyacetone phosphate to glyceraldehyde-3-phosphate [11]. Then, under the action of GADPH, PGK, PGAM1, ENO, PKs, etc., it was converted into pyruvate and generates ATP, and then finally lactate was generated under the action of lactate dehydrogenase. [9] Studies have revealed that the lactic acid produced by glucose through glycolytic metabolism in the process of cancer occurrence and development could provide energy for the proliferation, spread, and metastasis of cancer cells [27, 28], thereby inhibiting or blocking certain aspects of cancer cell glycolytic metabolism. Therefore, the key link, cutting off the energy supply

required for cell proliferation and migration, has become a hotspot in clinical research on tumor prognosis. For example, Jiang *et al.* [11] found that the *TPI1* gene could inhibit the occurrence of liver cancer, which suggested that *TPI1* can be used as a potential target for the treatment of liver cancer. Over the years, there have been many reports on the glycolytic metabolism pathway in the development of the reproductive system of male mammals, especially in the development of the testis and epididymis [7, 29, 30]. Studies in humans have found that the *TPI1* gene is expressed in the head of sperm, and anti-sperm antibodies can target *TPI1*, thereby preventing the sperm from undergoing acrosome reaction and preventing secondary binding of sperm to the zona pellucida [16]. Studies in adult bulls demonstrated that *TPI1* was expressed on the plasma membrane of the sperm acrosome [31]. Moreover, studies in mice have reported that *TPI1* was mainly expressed in the main segment of the sperm flagella, which is located in the tail of the sperm [32]. It is well known that oxidative phosphorylation provides energy for sperm movement [33]. Therefore, the *TPI1* gene may be mainly involved in the regulation of energy metabolism, so as to provide energy for normal movement of the sperm stored in the epididymis.

A previous study found that when the glycolytic metabolic process is abnormal, it could cause obstacles in sperm production, which can ultimately lead to male sterility [34]. Given that TPI1 could participate in the glycolytic metabolic pathway and the main metabolic mode of the SCs of the testis was glycolysis, the final product (lactic acid) produced through glycolysis provides nutrition for all levels of spermatogenic cells to ensure normal operation of the sperm formation process [6]. Studies have reported that dibromophenyl ether inhibits the glycolytic metabolic pathways in mouse testes and disrupts glucose metabolism homeostasis, thereby affecting the survival of germ cells [35]. Exogenous injection of 6-propyl-2-thiouracil could significantly reduce the glucose intake, LDH content, and lactic acid concentration in the testis of mice, ultimately increasing the apoptotic rate of germ cells and reducing the growth rate [36]. At the same time, when the expression of another important enzyme (PGAM1) in the glycolytic metabolic pathway was downregulated, the apoptosis rate of mouse SCs was significantly increased, the proliferation and migration abilities were restricted, and the proliferation of spermatogonia significantly inhibited cell apoptosis [37]. Herein, functional experiments confirmed that *TPI1* knockdown significantly inhibited activation of the glycolytic metabolism signaling pathways. Notably, previous studies have reported that the glycolytic metabolism of SCs has an important regulatory effect on the proliferation and apoptosis of spermatogenic cells [38, 39]. The results of this study found that overexpression of the *TPI1* gene could lead to increased pyruvate content, lactate production, ATP production, and LDH activity in SCs, whereas the expression of downstream genes in the glycolytic pathway was significantly upregulated. Consistently, this study showed that *TPI1* enhanced the synthesis of lactic acid during glycolytic metabolism of SCs, thereby enhancing the development of spermatogonial stem cells. These findings imply that *TPI1* may be involved in spermatogenesis, at least in part, by activating glycolytic metabolism signaling pathways to provide nutrients for spermatogenesis. The results also suggested that *TPI1* might participate in regulating the development and functional maintenance of sheep SCs through glycolytic metabolism to produce the energy substrate required for the development of spermatogenic cells-lactic acid. In addition, it was evident that *TPI1* participated in regulating the development of male sheep spermatogonial stem cells.

It has been reported that the miRNA regulatory network is effective and complex in physiological and pathological conditions [40]. The biological function of miRNAs is mainly mediated through pairing with a complementary site in the 3'UTR of the target mRNA, which results in post-transcriptional regulation by translational inhibition [41]. Accumulating evidence suggests that certain miRNAs participate in cancer progression by targeting distinct

mRNAs [20]. For example, miR-133b and miR-511 could regulate the expression of tumor suppressor genes to inhibit tumor growth [42]. MiR-200 could downregulate the expression of ZEB1 and ZEB2 in liver cancer, thereby inhibiting cell invasion and cancer progression [43]. Moreover, miR-1285-3p could inhibit the migration and invasion of cancer cells in liver cancer and pancreatic cancer [44, 45]. Numerous studies have shown that miRNAs participate in the pathogenesis of numerous diseases by regulating glycolytic metabolism and mitochondrial energy production [46, 47]. MiR-1285-3p inhibitor significantly improved the mitochondrial respiratory function and respiratory chain complex activity, thereby increasing ATP production, and reducing the number of cells with low MMP and mtROS to protect jejunal epithelial cells against Cu-induced toxic injury. Besides, miR-1285 mediated the deficiency in ATP generation by targeting AMPKα2 in immature boar SCs [45, 48]. The above results suggest that in addition to the role of miR-1285-3p in the occurrence of cancer, it has a certain regulatory effect on the growth and development of SCs.

To date, only a handful of reports have explored the role of miR-1285-3p in SCs glycolytic metabolism. In this study, analysis of multiple prediction websites identified *TPI1* as a functional target of miR-1285-3p (miR-1285-3p could bind to *TPI1* in a targeted manner). In addition, our dual luciferase reporter assay demonstrated that elevated expression of miR-1285-3p could significantly reduce the luciferase activity driven by the luciferase gene containing the 3'UTR of *TPI1*, which confirmed that miR-1285-3p can inhibit the expression of *TPI1* by directly targeting its 3'UTR. Furthermore, it was found that overexpression of *TPI1* weakened the inhibitory effect of miR-1285-3p on SCs apoptosis and the glycolytic metabolism signaling pathway. In summary, these findings identify miR-1285-3p as a new determinant of *TPI1* expression and establish a new miR-1285—3p/TPI1/glycolytic metabolism signaling pathway axis for spermatogenesis during testicular development in male animals.

## 5. Conclusion

In conclusion, our results further confirm that the *TPI1* gene regulates the glycolytic metabolic pathway of sheep SCs to provide energy substrates for the development of spermatogenic cells, thereby ensuring smooth progress of spermatogenesis. Mechanism investigations revealed that *TPI1* was the functional target of miR-1285-3p, and miR-1285-3p could reduce the expression of *TPI1* in Sertoli cells. In addition, the miR-1285-3p/TPI1 axis regulated the energy substrate required by spermatogenic cells, at least in part, by activating the glycolytic metabolism signaling pathway in Sertoli cells. Overall, these findings suggest that the miR-1285-3p/TPI1 axis provides a scientific basis for studying spermatogenesis and reproductive disorders in male mammals such as sheep.

## Supporting information

**S1 Table. Information of si-TPI1 sequence.**
(DOCX)

**S2 Table. Information of miRNA sequence.**
(DOCX)

**S3 Table. mRNA information of primer sequence.**
(DOCX)

**S4 Table. miRNA information of primer sequence.**
(DOCX)

**S1 Text. The original data and supplementary materials were stored in Datadryad at:**
**https://datadryad.org/stash/share/dg60cOPjYmG406-dwNzI2JdkOwleeknqaQI0nSR2oKA**
(**doi:10.5061/dryad.5dv41ns7k**).
(TXT)

## Acknowledgments

Thanks to all participants for their advice and support of this study.

## Author Contributions

**Conceptualization:** Xuejiao An, Youji Ma.

**Formal analysis:** Xuejiao An, Nana Chen.

**Funding acquisition:** Youji Ma.

**Methodology:** Xuejiao An, Taotao Li, Nana Chen, Manchun Su.

**Project administration:** Youji Ma.

**Software:** Xuejiao An, Taotao Li, Huihui Wang, Manchun Su, Huibin Shi, Xinming Duan.

**Supervision:** Youji Ma.

**Writing – original draft:** Xuejiao An.

**Writing – review & editing:** Youji Ma.

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
