## [Decision Letter · Decision Letter 0]

22 Dec 2021

PONE-D-21-36704miR-1285-3p targets TPI1 to regulate the glycolysis metabolism signaling pathway of Tibetan sheep Sertoli cellsPLOS ONE

Dear Dr. Ma,

Thank you for submitting your manuscript to PLOS ONE. After careful consideration, we feel that it has merit but does not fully meet PLOS ONE’s publication criteria as it currently stands. Therefore, we invite you to submit a revised version of the manuscript that addresses the points raised during the review process. There is a serious concern on the choice of miRNA (miR-1285-3p has 895 targets and TPI1 is not within the top 50 targets) and its effects on TP11 role in cell proliferation. Stronger evidence on the specificity is required. The transfection protocol details for experessing miR-1285-3p in Sertoli cells and the contribution of endogenously expressed miRNA needs to be provided. Manuscript should be written in an easily understandable format without any confusions. Figures and legends should provide data and description in a crystal clear format. Methodology details should be provided in detail. The manuscript should be revised by a native English speaker to avoid grammatical errors.

We look forward to receiving your revised manuscript.

Kind regards,

Suresh Yenugu

Academic Editor

PLOS ONE

Journal Requirements:

When submitting your revision, we need you to address these additional requirements. 1. Please ensure that your manuscript meets PLOS ONE's style requirements, including those for file naming. The PLOS ONE style templates can be found at https://journals.plos.org/plosone/s/file?id=wjVg/PLOSOne_formatting_sample_main_body.pdf and https://journals.plos.org/plosone/s/file?id=ba62/PLOSOne_formatting_sample_title_authors_affiliations.pdf 2. We noticed you have some minor occurrence of overlapping text with the following previous publication(s), which needs to be addressed: - https://doi.org/10.1016/j.gene.2021.145897 The text that needs to be addressed involves the Results section. In your revision ensure you cite all your sources (including your own works), and quote or rephrase any duplicated text outside the methods section. Further consideration is dependent on these concerns being addressed. 3. We suggest you thoroughly copyedit your manuscript for language usage, spelling, and grammar. If you do not know anyone who can help you do this, you may wish to consider employing a professional scientific editing service.  Whilst you may use any professional scientific editing service of your choice, PLOS has partnered with both American Journal Experts (AJE) and Editage to provide discounted services to PLOS authors. Both organizations have experience helping authors meet PLOS guidelines and can provide language editing, translation, manuscript formatting, and figure formatting to ensure your manuscript meets our submission guidelines. To take advantage of our partnership with AJE, visit the AJE website (http://learn.aje.com/plos/) for a 15% discount off AJE services. To take advantage of our partnership with Editage, visit the Editage website (www.editage.com) and enter referral code PLOSEDIT for a 15% discount off Editage services. If the PLOS editorial team finds any language issues in text that either AJE or Editage has edited, the service provider will re-edit the text for free. Upon resubmission, please provide the following:● The name of the colleague or the details of the professional service that edited your manuscript● A copy of your manuscript showing your changes by either highlighting them or using track changes (uploaded as a *supporting information* file)● A clean copy of the edited manuscript (uploaded as the new *manuscript* file) 4. To comply with PLOS ONE submissions requirements, please provide methods of sacrifice in the Methods section of your manuscript. 5. In your Methods section, please include a comment about the state of the animals following this research. Were they euthanized or housed for use in further research? If any animals were sacrificed by the authors, please include the method of euthanasia and describe any efforts that were undertaken to reduce animal suffering. 6. Thank you for stating the following financial disclosure:  "The study was supported by the Education science and technology innovation project of Gansu Province (GSSYLXM-02), National Natural Science Foundation of China (31960662), National Key R&D Program of China (2021YFD1100502) and "Innovation Star" project for outstanding graduate students of the Education Department of Gansu Province (2021CXZX-350)." Please state what role the funders took in the study.  If the funders had no role, please state: "The funders had no role in study design, data collection and analysis, decision to publish, or preparation of the manuscript." If this statement is not correct you must amend it as needed. Please include this amended Role of Funder statement in your cover letter; we will change the online submission form on your behalf. 
 7. Thank you for stating the following in your Competing Interests section:   "The authors declare that they have no known competing financial interests or personal relationships that could have appeared to influence the work reported in this paper." Please complete your Competing Interests on the online submission form to state any Competing Interests. If you have no competing interests, please state "The authors have declared that no competing interests exist.", as detailed online in our guide for authors at http://journals.plos.org/plosone/s/submit-now  This information should be included in your cover letter; we will change the online submission form on your behalf. 8. In your Data Availability statement, you have not specified where the minimal data set underlying the results described in your manuscript can be found. PLOS defines a study's minimal data set as the underlying data used to reach the conclusions drawn in the manuscript and any additional data required to replicate the reported study findings in their entirety. All PLOS journals require that the minimal data set be made fully available. For more information about our data policy, please see http://journals.plos.org/plosone/s/data-availability. Upon re-submitting your revised manuscript, please upload your study’s minimal underlying data set as either Supporting Information files or to a stable, public repository and include the relevant URLs, DOIs, or accession numbers within your revised cover letter. For a list of acceptable repositories, please see http://journals.plos.org/plosone/s/data-availability#loc-recommended-repositories. Any potentially identifying patient information must be fully anonymized. Important: If there are ethical or legal restrictions to sharing your data publicly, please explain these restrictions in detail. Please see our guidelines for more information on what we consider unacceptable restrictions to publicly sharing data: http://journals.plos.org/plosone/s/data-availability#loc-unacceptable-data-access-restrictions. Note that it is not acceptable for the authors to be the sole named individuals responsible for ensuring data access. We will update your Data Availability statement to reflect the information you provide in your cover letter. 9. PLOS ONE now requires that authors provide the original uncropped and unadjusted images underlying all blot or gel results reported in a submission’s figures or Supporting Information files. This policy and the journal’s other requirements for blot/gel reporting and figure preparation are described in detail at https://journals.plos.org/plosone/s/figures#loc-blot-and-gel-reporting-requirements and https://journals.plos.org/plosone/s/figures#loc-preparing-figures-from-image-files. When you submit your revised manuscript, please ensure that your figures adhere fully to these guidelines and provide the original underlying images for all blot or gel data reported in your submission. See the following link for instructions on providing the original image data: https://journals.plos.org/plosone/s/figures#loc-original-images-for-blots-and-gels.   In your cover letter, please note whether your blot/gel image data are in Supporting Information or posted at a public data repository, provide the repository URL if relevant, and provide specific details as to which raw blot/gel images, if any, are not available. Email us at plosone@plos.org if you have any questions. 10. PLOS requires an ORCID iD for the corresponding author in Editorial Manager on papers submitted after December 6th, 2016. Please ensure that you have an ORCID iD and that it is validated in Editorial Manager. To do this, go to ‘Update my Information’ (in the upper left-hand corner of the main menu), and click on the Fetch/Validate link next to the ORCID field. This will take you to the ORCID site and allow you to create a new iD or authenticate a pre-existing iD in Editorial Manager. Please see the following video for instructions on linking an ORCID iD to your Editorial Manager account: https://www.youtube.com/watch?v=_xcclfuvtxQ

Reviewers' comments:

Reviewer's Responses to Questions

**Comments to the Author**

1. Is the manuscript technically sound, and do the data support the conclusions?

Reviewer #1: Partly

Reviewer #2: Partly

2. Has the statistical analysis been performed appropriately and rigorously? 

Reviewer #1: Yes

Reviewer #2: Yes

3. Have the authors made all data underlying the findings in their manuscript fully available?

Reviewer #1: Yes

Reviewer #2: No

4. Is the manuscript presented in an intelligible fashion and written in standard English?

Reviewer #1: No

Reviewer #2: Yes

5. Review Comments to the Author

Reviewer #1: Comments:

An et al. studied the role of Triose phosphate isomerase 1 (TPI1) in testicular Sertoli cells (Sc) isolated from Tibetan sheep. The authors demonstrate that TPI1 regulates Sc proliferation, apoptosis, and gene expression. Authors show that mir-1285-3p directly targets TPI1, and manipulating the levels of the microRNA has a similar effect that is observed upon knockdown/over-expression of TPI1 in Sc. This is an interesting manuscript; however, some aspects of the study need to be clarified, and more experiments are necessary to describe the manuscript's findings. Also, serious issues with the writing must be addressed. Overall, the paper feels premature and would benefit from some major rewriting. Throughout the article, including abstract, many vocabulary, grammar, and pose errors make it very hard to follow and interpret.

Major comments:

The abstract is poorly written.

All the figure legends should be revised—for example, Fig2: Changing of TPI1 protein in SCs after silence or overexpression.

What about the images or graphs in the figure? Don’t they need any explanation? What about the statistical significance? What methods were used? Fig 2 is cited at the end of the paragraph. Very hard to understand.

In the text, it was mentioned si-PGAM1, whereas in the figure TPI1.

In the 3rd figure and 3.3, what is CCK8? How can first-time readers understand?

Difference in overexpression or silence after 24 h of transection: What is transection?

The number of cells in the si-PTPI1 group was significantly lower than that in the NC group.

What is si-PTTPI1?

Further qRT-PCR was used to detect the changes in the expression of proliferation, cycle and apoptosis-related genes at the mRNA level.

What is cycle?

The above typos are from a single paragraph. I haven’t included the English corrections! Similarly, every paragraph has typos and English corrections.

1. Materials and methods (Isolation and culture of Sertoli cells): The authors are suggested to provide the culture purity data (GATA4/SOX9 immunostaining) and the expression of TPI1 (immunostaining, immunoblotting) in cultured Sc in the main figure. This is important as contamination by other testicular cell types (also known to express TPI1) needs to be ruled out.

2. For how many days were the cells cultured in vitro? After how many days were the cells transfected, and what was the transfection efficiency? The authors should provide information regarding the same in the material and methods section.

3. The authors have studied the effect of TPI1 on Sc proliferation and apoptosis. There is a dogma that Sc proliferates during the neonatal infantile period and stops proliferating onset of puberty. Thus authors are suggested to check the expression of TPI1 changes during pubertal maturation of Sc. Along the same lines, does the expression of mi-1285-3p change during the functional maturation of Sc at puberty? Incorporation of this data would improve the quality of the manuscript.

4. How does knock-down/over-expression of TPI1 regulate the mRNA levels of glycolytic pathway genes? Does TPI1 regulate the expression of these genes directly, or is the change in gene expression indirect (due to a shift in metabolic flux as a consequence of TPI1 knockdown/over-expression)? The authors should comment on this in the discussion section.

5. The authors are suggested to provide separate figures for the TPI1 knockdown and over-expression experiments along with the consequent effect(s) of knockdown/over-expression of TPI1 on Sc function (proliferation, apoptosis, gene expression etc..).

6. Which gene was used to normalize the gene expression data? This needs to be mentioned in the materials and methods section.

7. Results section 3.4- the authors state that “the expression of ENO1, PKM, LDHA, LDHB, MCT1, GAPDH, and PGK genes were significantly down-regulated in the si-PGAM1 group (P<0.05), indicating the over-expression of the PGAM1 gene could increase the expression of downstream genes while silencing the PGAM1 gene could reduce the expression of downstream genes”. PGAM1 needs to be replaced with TPI1 in the text.

8. There are major grammatical errors and awkward sentences in the manuscript which need to be corrected.

Minor:

• Line 2. Repetition of sentences. breeds in China; Tibetan sheep (Ovis aries) is one of the three major coarse-haired sheep breeds in China. breeds in China.

• Page no-8 In the initial introduction, the importance of spermatogenesis should be brief and crisp. It covers half of the introduction with repeated sense.

• The authors should provide details of Ab used for isolation, polyclonal or monoclonal?

• Would the authors mind providing precise details of the sample collection procedure (collected from a slaughterhouse or by sacrifice) and the number of testicles collected for each experiment?

• No need to rewrite the material method in the result section.

• The author should provide good-quality images. Fig 5 and 6 captions are not readable.

Reviewer #2: In this study, the authors suggest a regulatory mechanism of Sertoli Cell metabolism and function based on the control of TPI1 expression via the miRNA miR-1285-3p. To support their hypothesis, the authors have designed an in vitro model based in primary Sertoli cells of Tibetan sheep. The manuscript is fairly well-written, although some methodological aspects are not easy the follow. The authors provide evidence of this regulatory mechanism at both gene and protein expression levels. Despite that, I have some reserve about the causal relationship between miR-1285-3p, TPI1 and the effects on Sertoli cell proliferation and downstream gene expression observed by the authors, due to the multiple targets of the miR-1285-3p.

Major issues:

1) It was not possible to confirm the existence of a binding site to miR-1285-3p in the TPI1 gene using miRanda, because the website is no longer available. However, using miRDB (http://mirdb.org) it is possible to estimate two binding sites between TPI1 and miR-1285-3p in 3’ UTR, in humans. However, according to this database, there are 41 miRNAs targeting the TPI1 gene, and the miR-1285-3p is not the miRNA with higher “score”. The authors state in the introduction that this miRNA was selected because it is the most expressed in the Tibetan Sheep among the miRNAs with high affinity for the TPI1 gene. Yet, miR-1285-3p has 895 targets and TPI1 is not within the top 50 targets. Among the miR-1285-3p targets with higher score than TPI1 are several genes that could be involved in cell proliferation and metabolic regulation. Therefore, it is unlikely that the action of miR-1285-3p in Sertoli Cells is restricted to the regulation of TPI1 gene. To claim that, the authors would have to characterise the transcriptome of the Sertoli Cells transfected with the TPI1 inhibitor.

2) Related to the previous issue, the authors do not refer the parameters to predict miRNAs targeting TPI1 using miRanda, especially which animal database was used. Although the primers were specific to Tibetan Sheep, the target prediction was likely performed using a database for another species. This limitation must be discussed.

3) It is not clear to me how the miR-1285-3p mimic is expressed in Sertoli Cells by means of a plasmid, and the supplementary files are not available. How can the authors guarantee that the sequence will mimic the behaviour of the original miR-1285-3p? If the mimic sequence is expressed along the luciferase gene, then it must be processed post-transcriptionally. This must also be discussed by the authors.

4) In section 3.5 and Figure 5, the authors mention the miR-3614-5p. Did the authors mean miR-1285-3p?

Minor issues:

1) Please replace the term “testicles” by the term “testes” (singular “testis”).

2) QRT-PCR -> qRT-PCR

3) Multipanel figures must be labelled uniformly. In some figures the panels are ordered from left to right, top to bottom, but in other figures the authors label from top to bottom and only then from left to right. Although I understand the authors have a lot of information to show, please consider relegating some figure panels to supplementary data to improve the visualization of main manuscript figures.

6. PLOS authors have the option to publish the peer review history of their article (what does this mean?). If published, this will include your full peer review and any attached files.

Reviewer #1: No

Reviewer #2: **Yes: **Luís Crisóstomo

---

## [Author Response · Author response to Decision Letter 0]

1 Apr 2022

Response to Reviewer 1 Comments

We thank the reviewer for the time in closely viewing our manuscript. Most of the suggestions have been adopted in the revision.

Major comments: 

Point 1: The abstract is poorly written.

Response 1: Thanks for your comments, we have revised the abstract section, and the specific revision results are as follows: Glycolysis in Sertoli cells (SCs) can provide energy substrates for the development of spermatogenic cells. Triose phosphate isomerase 1 (TPI1) is one of the key catalytic enzymes involved in glycolysis. However, the biological function of TPI1 in SCs and its role in glycolytic metabolic pathways are poorly understood. On the basis of a previous research, we isolated primary SCs from Tibetan sheep, and overexpressed TPI1 gene to determine its effect on the proliferation, glycolysis, and apoptosis of SCs. Secondly, we investigated the relationship between TPI1 and miR-1285-3p, and whether miR-1285-3p regulates the proliferation and apoptosis of SCs, and participates in glycolysis by targeting TPI1. Results showed that overexpression of TPI1 increased the proliferation rate and decreased apoptosis of SCs. In addition, overexpression of TPI1 altered glycolysis and metabolism signaling pathways and significantly increased amount of the final product lactic acid. Further analysis showed that miR-1285-3p inhibited TPI1 by directly targeting its 3'untranslated region. Overexpression of miR-1285-3p suppressed the proliferation of SCs, and this effect was partially reversed by restoration of TPI1 expression. In summary, this study shows that the miR-1285-3p/TPI1 axis regulates glycolysis in SCs. These findings add to our understanding on the regulation of spermatogenesis in sheep and other mammals.

Point 2: All the figure legends should be revised—for example, Fig2: Changing of TPI1 protein in SCs after silence or overexpression. What about the images or graphs in the figure? Don’t they need any explanation? What about the statistical significance? What methods were used? Fig 2 is cited at the end of the paragraph. Very hard to understand.

Response 2: Thanks for your comments, I have modified it in the text.

Point 3: In the text, it was mentioned si-PGAM1, whereas in the figure TPI1.

Response 3: Thanks for your comments, this is a mistake in my negligent text, it should be si-TPI1, I have made changes in the text.

Point 4: In the 3rd figure and 3.3, what is CCK8? How can first-time readers understand?

Response 4: Thanks for your comments, CCK8 is Cell counting kit-8, I have modified it in the text.

Point 5: Difference in overexpression or silence after 24 h of transection: What is transection? The number of cells in the si-PTPI1 group was significantly lower than that in the NC group. What is si-PTTPI1? Further qRT-PCR was used to detect the changes in the expression of proliferation, cycle and apoptosis-related genes at the mRNA level. What is cycle? The above typos are from a single paragraph. I haven’t included the English corrections! Similarly, every paragraph has typos and English corrections.

Response 5: Thanks for your comments, I have double checked the article and corrected the mistakes.

Point 6: Materials and methods (Isolation and culture of Sertoli cells): The authors are suggested to provide the culture purity data (GATA4/SOX9 immunostaining) and the expression of TPI1 (immunostaining, immunoblotting) in cultured Sc in the main figure. This is important as contamination by other testicular cell types (also known to express TPI1) needs to be ruled out.

Response 6: Thanks for your comments, I have modified the results section, and the specific modification results are as follows:

3.1 Identification of Tibetan sheep primary SCs

Immunofluorescence staining of SCs with specific antibody GATA4 and TPI1 antibody showed that purified SCs could synthesize and express SCs-specific binding protein GATA4, and staining with TPI1 antibody showed that TPI1 protein was expressed in SCs. The results could rule out contamination by other testis cell types, and the purified SCs could be used in subsequent experiments (Fig. 1).

Fig. 1 Immunofluorescence staining of primary SCs of Tibetan sheep (20×).

Point 7: For how many days were the cells cultured in vitro? After how many days were the cells transfected, and what was the transfection efficiency? The authors should provide information regarding the same in the material and methods section.

Response 7: Thanks for your comments, SCs were cultured in vitro and purified to the F5 generation for transfection, and the cells at 24, 48 and 72 after transfection were collected, and the transfection efficiency was detected by fluorescence qRT-PCR. It was found that the transfection efficiency at 48h was the highest. Therefore, other follow-up experiments were selected 48h after transfection. I have modified it in the article.

Point 8: The authors have studied the effect of TPI1 on Sc proliferation and apoptosis. There is a dogma that Sc proliferates during the neonatal infantile period and stops proliferating onset of puberty. Thus authors are suggested to check the expression of TPI1 changes during pubertal maturation of Sc. Along the same lines, does the expression of mi-1285-3p change during the functional maturation of Sc at puberty? Incorporation of this data would improve the quality of the manuscript.

Response 8: Thanks for your comments, our research group tested the expression of TPI1 in the testis of Tibetan sheep at different developmental stages, and found that the expression of its protein and mRNA before sexual maturity (3 mouth) was significantly lower than that of sexual maturity (1 year) and somatic maturity (3 years), and stabilized after sexual maturity.

Point 9: How does knock-down/over-expression of TPI1 regulate the mRNA levels of glycolytic pathway genes? Does TPI1 regulate the expression of these genes directly, or is the change in gene expression indirect (due to a shift in metabolic flux as a consequence of TPI1 knockdown/over-expression)? The authors should comment on this in the discussion section.

Response 9: Thanks for your comments, as one of the key enzymatic molecules in the glycolytic pathway, TPI1 catalyzes the conversion of dihydroxyacetone phosphate to glyceraldehyde-3-phosphate [11]. Then, under the action of GADPH, PGK, PGAM1, ENO, PKs, etc., it is converted into pyruvate and generates ATP, and then finally lactate is generated under the action of lactate dehydrogenase [9]. Therefore, changes in TPI1 gene expression can directly change the changes in the downstream genes of the glycolytic pathway, and through these genes regulate the changes of enzymes in the glycolytic metabolic pathway, and finally change the content of its final product, lactate.

Point 10: The authors are suggested to provide separate figures for the TPI1 knockdown and over-expression experiments along with the consequent effect(s) of knockdown/over-expression of TPI1 on Sc function (proliferation, apoptosis, gene expression etc..).

Response 10: Thanks for your comments, the first half of the article is the test of TPI1 gene knockdown or overexpression, and also explores the effect of TPI1 gene knockdown or overexpression on SCs proliferation, apoptosis and glycolysis metabolic pathways.

Point 11: Which gene was used to normalize the gene expression data? This needs to be mentioned in the materials and methods section.

Response 11: Thanks for your comments, β-actin gene was used to normalize the gene expression data, I have added in the methods section of the text.

Point 12: Results section 3.4- the authors state that “the expression of ENO1, PKM, LDHA, LDHB, MCT1, GAPDH, and PGK genes were significantly down-regulated in the si-PGAM1 group (P<0.05), indicating the over-expression of the PGAM1 gene could increase the expression of downstream genes while silencing the PGAM1 gene could reduce the expression of downstream genes”. PGAM1 needs to be replaced with TPI1 in the text.

Response 12: Thanks for your comments, I have changed “PGAM1” to “TPI1”in the text.

Point 13: There are major grammatical errors and awkward sentences in the manuscript which need to be corrected.

Response 13: Thanks for your comments, I have found native English speakers to revise the grammar and sentences.

Minor:

Point 1: Line 2. Repetition of sentences. breeds in China; Tibetan sheep (Ovis aries) is one of the three major coarse-haired sheep breeds in China. breeds in China.

Response 1: Thanks for your comments, I have modified it in the text.

Point 2: Page no-8 In the initial introduction, the importance of spermatogenesis should be brief and crisp. It covers half of the introduction with repeated sense.

Response 2: Thanks for your comments, I have modified it in the text.

Point 3: The authors should provide details of Ab used for isolation, polyclonal or monoclonal?

Response 3: Thanks for your comments, the construction of the vector was completed by Genewiz Biological. We just verified the double-enzyme digestion of the overexpression vector synthesized by them, so the single-clonal and polyclonal information was not described in detail in the method.

Point 4: Would the authors mind providing precise details of the sample collection procedure (collected from a slaughterhouse or by sacrifice) and the number of testicles collected for each experiment?

Response 4: Thanks for your comments, three well-grown 3-month-old Tibetan sheep were selected and brought back to the laboratory for intravenous injection of sodium pentobarbital, no heartbeat, continuous involuntary breathing for 2-3 min, no blink reflex, and then bilateral testes tissues were collected. I have modified it in methods section.

Point 5: No need to rewrite the material method in the result section.

Response 5: Thanks for your comments, I have modified the results section.

Point 6: The author should provide good-quality images. Fig 5 and 6 captions are not readable.

Response 6: Thanks for your comments, I have modified the Fig 5 and 6.

Response to Reviewer 2 Comments

We thank the reviewer for the time in closely viewing our manuscript. Most of the suggestions have been adopted in the revision.

Major issues:

Point 1: It was not possible to confirm the existence of a binding site to miR-1285-3p in the TPI1 gene using miRanda, because the website is no longer available. However, using miRDB (http://mirdb.org) it is possible to estimate two binding sites between TPI1 and miR-1285-3p in 3’ UTR, in humans. However, according to this database, there are 41 miRNAs targeting the TPI1 gene, and the miR-1285-3p is not the miRNA with higher “score”. The authors state in the introduction that this miRNA was selected because it is the most expressed in the Tibetan Sheep among the miRNAs with high affinity for the TPI1 gene. Yet, miR-1285-3p has 895 targets and TPI1 is not within the top 50 targets. Among the miR-1285-3p targets with higher score than TPI1 are several genes that could be involved in cell proliferation and metabolic regulation. Therefore, it is unlikely that the action of miR-1285-3p in Sertoli Cells is restricted to the regulation of TPI1 gene. To claim that, the authors would have to characterise the transcriptome of the Sertoli Cells transfected with the TPI1 inhibitor.

Response 1: Thanks for your comments, in our earlier studies, we explored the proteomics of sheep testis tissue and found that the TPI1 protein was expressed in sheep testis tissue. After conducting further functional annotations, we found that the TPI1 protein participates in the glycolytic metabolism pathway. In order to further verify its role in testicular development of Tibetan sheep, we further explored the expression and localization of TPI1 in testis tissue of Tibetan sheep at different developmental stages (prepubertal, sexual maturity and adulthood), and found that its mRNA and protein expression varied with the It increases with age and is mainly located in SCs. However, its role in SCs and its regulation of glycolytic metabolism have not been reported yet. This study mainly explored the role of TPI1 in the glycolysis process of Tibetan sheep SCs on the basis of previous studies. Since most of the previous research on TPI1 was in cancer, some reports on the regulation of miRNA on TPI1 were found in the literature when reviewing the literature, so combined with previous reports and prediction software (miRDB(http://www.mirdb.org) and Targetscan (http://www.targetscan.org), screened out several miRNAs that have a targeting relationship with TPI1, and found that miR-1285-3p had the highest expression in Sertoli cells, and the dual luciferase reporter gene assay was used to detect the high expression of miR-1285-3p. The targeting relationship between the three miRNAs and TPI1 was verified, and it was found that only miR-1285-3p had a targeting relationship, so this study directly selected miR-1285-3p for research.

Point 2: Related to the previous issue, the authors do not refer the parameters to predict miRNAs targeting TPI1 using miRanda, especially which animal database was used. Although the primers were specific to Tibetan Sheep, the target prediction was likely performed using a database for another species. This limitation must be discussed.

Response 2: Thanks for your comments, in this study, the first prediction was based on the human database through miRDB. After predicting that miR-1285-3p has a targeting relationship with TPI1, the sequence of sheep miR-1285-3p was downloaded through miRNAbase. The seed region of miR-1285-3p was aligned with the 3'UTR region of the ovine TPI1 gene, and it was found that it was completely complementary. Therefore, it was shown that TPI1 also has a targeting relationship with miR-1285-3p in sheep, so follow-up experiments were carried out.

Point 3: It is not clear to me how the miR-1285-3p mimic is expressed in Sertoli Cells by means of a plasmid, and the supplementary files are not available. How can the authors guarantee that the sequence will mimic the behaviour of the original miR-1285-3p? If the mimic sequence is expressed along the luciferase gene, then it must be processed post-transcriptionally. This must also be discussed by the authors.

Response 3: Thanks for your comments, the mimics and inhibitors are designed according to the gene sequence on the miRNAbase. The sense strand of the mimics sequence is the miRNA sequence, and the antisense strand is the sequence of the sense strand after removing the reverse complement of the last two bases and adding UU.The inhibitor sequence is the sequence after the complete reverse complementation of the base sequence. Synthetic miRNA mimics are chemical compounds of more than 20 bases, which simulate endogenous gene sequences through transfection with transfection reagents and then perform gene expression or knockdown.

Point 4: In section 3.5 and Figure 5, the authors mention the miR-3614-5p. Did the authors mean miR-1285-3p?

Response 4: Thanks for your comments, I made a typo, it should be miR-1285-3p, I have made changes in the text.

Minor issues:

Point 1: Please replace the term “testicles” by the term “testes” (singular “testis”).

Response 1: Thanks for your comments, I have modified it in the article.

Point 2: QRT-PCR -> qRT-PCR

Response 2: Thanks for your comments, I have modified it in the article.

Point 3: Multipanel figures must be labelled uniformly. In some figures the panels are ordered from left to right, top to bottom, but in other figures the authors label from top to bottom and only then from left to right. Although I understand the authors have a lot of information to show, please consider relegating some figure panels to supplementary data to improve the visualization of main manuscript figures.

Response 3: Thanks for your comments, I have modified the figures in the article.

---

## [Decision Letter · Decision Letter 1]

20 Apr 2022

PONE-D-21-36704R1miR-1285-3p targets TPI1 to regulate the glycolysis metabolism signaling pathway of Tibetan sheep Sertoli cellsPLOS ONE

Dear Dr. Ma,

Thank you for submitting your manuscript to PLOS ONE. After careful consideration, we feel that it has merit but does not fully meet PLOS ONE’s publication criteria as it currently stands. Therefore, we invite you to submit a revised version of the manuscript that addresses the points raised during the review process.

Specifically:Provide the details of the databases used and appropriately quote your previous works that pertain to this study.

We look forward to receiving your revised manuscript.

Kind regards,

Suresh Yenugu

Academic Editor

PLOS ONE

Journal Requirements:

Reviewers' comments:

Reviewer's Responses to Questions

**Comments to the Author**

1. If the authors have adequately addressed your comments raised in a previous round of review and you feel that this manuscript is now acceptable for publication, you may indicate that here to bypass the “Comments to the Author” section, enter your conflict of interest statement in the “Confidential to Editor” section, and submit your "Accept" recommendation.

Reviewer #1: All comments have been addressed

Reviewer #2: (No Response)

2. Is the manuscript technically sound, and do the data support the conclusions?

Reviewer #1: Yes

Reviewer #2: Yes

3. Has the statistical analysis been performed appropriately and rigorously? 

Reviewer #1: Yes

Reviewer #2: Yes

4. Have the authors made all data underlying the findings in their manuscript fully available?

Reviewer #1: Yes

Reviewer #2: Yes

5. Is the manuscript presented in an intelligible fashion and written in standard English?

Reviewer #1: Yes

Reviewer #2: Yes

6. Review Comments to the Author

Reviewer #1: (No Response)

Reviewer #2: This version of the manuscript is significantly better than the original submission. Most of the comments were addressed by the authors, making the manuscript more organized, clear, and scientifically sound. Personally, I am satisfied with the replies of the authors to my comments, but I consider that the authors have not properly included their justifications in the manuscript.

Particularly, it is important to state in the "Methods" the parameters of the miRNA target prediction tool, especially the species database. For instance, miRDB does not have a sheep database, so I guess the authors have relied on another database. Then, this issue must be discussed as a limitation of the study.

Also, I cannot find in the text any reference about the author's previous work, notably the claim that miR-1285-3p is expressed the most in testis of Tibetan sheep. This previous work is important to support the rationale for the present study, especially due to the lack of information and databases in the species.

There is a erroneous reference to Figure 5 in line 335.

7. PLOS authors have the option to publish the peer review history of their article (what does this mean?). If published, this will include your full peer review and any attached files.

Reviewer #1: No

Reviewer #2: **Yes: **Luís Crisóstomo

---

## [Author Response · Author response to Decision Letter 1]

15 May 2022

Response to Reviewer Comments

We thank the reviewer for the time in closely viewing our manuscript. Most of the suggestions have been adopted in the revision.

Point 1: Particularly, it is important to state in the "Methods" the parameters of the miRNA target prediction tool, especially the species database. For instance, miRDB does not have a sheep database, so I guess the authors have relied on another database. Then, this issue must be discussed as a limitation of the study.

Response 1: Thanks for your comments, in this study, the first prediction was based on the human database through miRDB. It was found that 6 miRNAs had a targeting relationship with TPI1. Then the miRNA sequences of sheep were downloaded by miRNAbase software, and the sequences were compared, and it was found that their seed regions were the same. Then primers were designed to detect the expression of these 6 miRNAs in Tibetan sheep SCs. It was found that the expression level of miR-1285-3p was the highest in Tibetan sheep, and then the targeting relationship between miR-1285-3p and TPI1 was further predicted. It was found that the seed region of miR-1285-3p and the 3'UTR of TPI1 gene could be completely complementary. Therefore, it showed that miR-1285-3p had a targeting relationship with TPI1, and we further verified the targeting relationship with the double luciferase reporter gene.

Point 2: Also, I cannot find in the text any reference about the author's previous work, notably the claim that miR-1285-3p is expressed the most in testis of Tibetan sheep. This previous work is important to support the rationale for the present study, especially due to the lack of information and databases in the species.

Response 2: Thanks for your comments, the result that miR-1285-3p was most expressed in Tibetan sheep SCs was originally included in Figure S1 as supplementary information. According to expert advice, I have put this part of the results in 3.5 and put the picture in Fig. 6F. As shown below:

We studied the expression of 6 miRNAs predicted to be targeted to TPI1 in SCs and found that the expression level of miR-1285-3p in SCs was significantly higher than that of other miRNAs(P<0.05) (Fig. 6F).

 Fig. 6 TPI1 expression is downregulated by miR-1285-3p directly targeting of the 3’-UTR of TPI1. A, TPI1 was found for the potential regulatory targets of miR-1285-3p using prediction tool. B, Dual-luciferase reporter assay analysis of the target relationship of miR-1285-3p with TPI1. F, Expression of miRNA in SCs. The TPI1 expression levels were determined by real-time PCR analysis (D) western blot (H) after transfection with the miR-1285-3p mimics or negative control or after transfection with the miR-1285-3p inhibitor or negative control in SCs. CCK-8 (E), Flow cytometry (G) and the expression of proliferation, cycle and apoptosis related genes (C) was used to detect the apoptosis of SCs after transfection miR-1285-3p mimics, negative control, miR-1285-3p inhibitor or negative control. The expression levels of downstream genes (J) and the content of glycolytic metabolic pathway products, energy and key enzymes (I) of SCs after transfection miR-1285-3p mimics, negative control, miR-1285-3p inhibitor or negative control.

Point 3: There is a erroneous reference to Figure 5 in line 335.

Response 3: Thanks for your comments, I have modified it in the text.

---

## [Decision Letter · Decision Letter 2]

9 Jun 2022

miR-1285-3p targets TPI1 to regulate the glycolysis metabolism signaling pathway of Tibetan sheep Sertoli cells

PONE-D-21-36704R2

Dear Dr. Ma,

We’re pleased to inform you that your manuscript has been judged scientifically suitable for publication and will be formally accepted for publication once it meets all outstanding technical requirements.

Kind regards,

Suresh Yenugu

Academic Editor

PLOS ONE

Additional Editor Comments (optional):

Reviewers' comments:

Reviewer's Responses to Questions

**Comments to the Author**

1. If the authors have adequately addressed your comments raised in a previous round of review and you feel that this manuscript is now acceptable for publication, you may indicate that here to bypass the “Comments to the Author” section, enter your conflict of interest statement in the “Confidential to Editor” section, and submit your "Accept" recommendation.

Reviewer #2: All comments have been addressed

2. Is the manuscript technically sound, and do the data support the conclusions?

Reviewer #2: Yes

3. Has the statistical analysis been performed appropriately and rigorously? 

Reviewer #2: Yes

4. Have the authors made all data underlying the findings in their manuscript fully available?

Reviewer #2: Yes

5. Is the manuscript presented in an intelligible fashion and written in standard English?

Reviewer #2: Yes

6. Review Comments to the Author

Reviewer #2: (No Response)

7. PLOS authors have the option to publish the peer review history of their article (what does this mean?). If published, this will include your full peer review and any attached files.

Reviewer #2: **Yes: **Luís Crisóstomo

---

## [Editor Report · Acceptance letter]

14 Sep 2022

PONE-D-21-36704R2 

miR-1285-3p targets TPI1 to regulate the glycolysis metabolism signaling pathway of Tibetan sheep Sertoli cells 

Dear Dr. Ma:

I'm pleased to inform you that your manuscript has been deemed suitable for publication in PLOS ONE. Congratulations! Your manuscript is now with our production department. 

Kind regards, 

on behalf of

Dr. Suresh Yenugu 

Academic Editor

PLOS ONE